# Barriers and Facilitators of Access to Healthcare Among Immigrants with Disabilities: A Qualitative Meta-Synthesis

**DOI:** 10.3390/healthcare13030313

**Published:** 2025-02-04

**Authors:** Ponsiano Ngondwe, Gashaye Melaku Tefera

**Affiliations:** College of Social Work, Florida State University, Tallahassee, FL 32304, USA; gtefera@fsu.edu

**Keywords:** immigrants with disabilities, access to healthcare, barriers of access, facilitators of access

## Abstract

Background: Immigrants with disabilities (IWDs) are disproportionately affected by a lack of access to healthcare services and face unique challenges compared to the general population. This qualitative meta-synthesis examines the barriers, facilitators, and lived experiences of IWDs accessing healthcare in the U.S. and Canada. Methods: A theory-generating qualitative meta-synthesis approach was used to analyze and synthesize raw qualitative data. Using eight databases, 752 studies were retrieved, and 10 were selected and synthesized after a three-stage review. The final articles were assessed using the Critical Appraisal Skills Program (CASP) checklist, and a PRISMA flow chart was used to report on the selection process. Results: The analysis identified structural barriers, including the bureaucracy and complexity of the system, healthcare costs, transportation, communication, long wait times, and a lack of integrated services. Cultural barriers included denial and trust, stigma and discrimination, awareness and language gaps, and lack of social support. Facilitators of access included support from immediate family members, community health centers, and social workers. Conclusions: The findings highlight the need for policy reforms to reduce bureaucratic hurdles, improve communication within healthcare systems, and enhance cultural competence among healthcare providers. Addressing these issues through integrated service models and targeted support can significantly improve the quality of life as a result of improved healthcare access for IWDs.

## 1. Introduction

The term immigrant is commonly used to represent people who live in countries outside those where they were born [1]. This study relates the case of immigrants who moved from other countries to live in the United States and Canada, and the term disability is defined as a long-term physical, mental, intellectual, or sensory impairment that may hinder an individual’s full participation in society [2]. This denotes the adverse aspects of the interaction between a person’s health condition(s) and their contextual factors, including environmental and personal elements [3,4]. Although the exact number of IWDs is not known, there is indication from different studies that disability is significantly present among the immigrant population. For example, researchers at Urban Institute reported that in 2022, 5.6% of the immigrant population between 18 and 64 had a disability [4].

In the Canadian context, the 2016 Census indicated that 13.7% of immigrants reported having at least one disability, highlighting the significant presence of disabilities within the immigrant population in Canada as well [3]. Another American Community Survey carried out over a five-year period (2015–2019), showed that 5.6% of nonelderly immigrants have a disability, with the highest being among the Black Latinx immigrants at 10.2% and the lowest for non-Latinx Asian immigrants at 4.2% [4]. Similar patterns are observed in Canada, where immigrants from certain regions, such as Africa and the Middle East, report higher disability rates compared to other immigrant groups [3]. This is supported by further research showing that IWDs represent a unique and often marginalized group, combining the challenges associated with both immigration and disability [5].

Immigrants with disabilities encounter multifaceted barriers, including linguistic challenges, cultural differences, and limited access to resources, which exacerbate their disability-related challenges [6]. Research has further revealed that IWDs are more likely to experience poverty, unemployment, and lower levels of educational attainment compared to both immigrants without disabilities and native-born individuals with disabilities [4]. This demographic’s unique situation necessitates a nuanced understanding of their experiences to inform policy and practice. In Canada, IWDs also face similar barriers, and studies have shown that they experience higher rates of unemployment and poverty compared to both immigrants without disabilities and native-born Canadians with disabilities [4]. When it comes to healthcare, IWDs in the U.S. face significant disparities in accessing healthcare services because of a combination of factors such as legal status, limited financial capacity, lack of employment-based insurance coverage, and a lack of familiarity with the healthcare system [7]. A previous study documented that IWDs suffer from inadequate health insurance and difficulties navigating the complex service system [8]. Immigrants with no insurance are likely to forego preventive services or needed medical assistance, which will further predispose them to more complex disability-related problems [9]. Compared to people without disabilities, IWDs are more likely to be affected by physical inactivity, obesity, and harmful health behaviors such as smoking and alcohol use [10]. In addition, ethnicity-based health beliefs and health misinformation significantly influence healthcare-seeking behavior and decision-making among IWDs and their families [9]. Canadian IWDs face similar healthcare access issues, compounded by challenges such as long wait times and a lack of culturally appropriate care [11]. Most concerning, service provision for IWDs is very limited and riddled with several gaps, particularly in areas such as language accessibility, cultural competence, and specialized care [12]. Studies have highlighted the inadequacy of current services, including the lack of interpreter services in healthcare settings and the limited reach of disability support services within immigrant communities [6]. Research also showed that most providers have limited knowledge of disability rights and use a narrow biomedical perspective of disability that ignores the socio-cultural and environmental aspects of immigrants’ disability issues [12]. A disconnection between disability service organizations and immigrant service organizations is also another challenge in expanding healthcare for IWDs [12,13]. These service gaps impede access to healthcare and affect the overall quality of life and social integration. The situation also results in increased risks of chronic diseases, mental health issues, and overall lower well-being in this population [4]. Furthermore, the absence of appropriate support systems can lead to social isolation and economic hardships, compounding the challenges faced by IWDs [14]. Both the U.S. and Canada still face challenges in integrating services for immigrants and individuals with disabilities, significantly impacting their overall health and well-being [11].

Despite the increased vulnerabilities of the population, the representation of IWDs in the immigrant health literature remains sparse. Particularly, there is no known comprehensive information that synthesizes and reports on the barriers to healthcare access among this sub-population. Although there are studies that independently reported on disability among immigrants in particular groups, such as Asian or Latino immigrants, most of these studies relied on national surveys and a limited number of variables [10,15]. There are no comprehensive reports that delve into the lived experiences of IWDs and provide nuanced discussions on their healthcare experience to help develop policies and interventions tailored to this group’s needs. This qualitative meta-synthesis aims to address the above gaps, particularly concerning the intersectionality of immigration status and disability concerning healthcare access, by analyzing qualitative studies that documented lived experiences [16]. As a theory-generating meta-synthesis [17] this study also aims to develop an inductively developed comprehensive framework that helps us to better understand barriers to access and serves as a blueprint for future large-scale studies that aim to address the complex factors involved in impeding access among IWDs. This study answers the following research question:

What are the barriers and facilitators of access to healthcare experienced by immigrants with disabilities in the U.S. and Canada? Since this study analyzed research articles from two different countries, a brief background on the differences and shared challenges of the two healthcare systems is presented below. Despite the differences in the two healthcare systems, the challenges and facilitators experienced by IWDs reported in the synthesized studies are predominantly overlapping, and with a few unique challenges, as we acknowledged in the discussion and limitation sections.

## 2. U.S. and Canadian Healthcare Systems

Although this study focused on synthesizing studies from U.S. and Canada due to limited number of publications on the topic of interest, we acknowledge that the healthcare systems of the two countries differ in structure, funding, and access to services [18]. In the United States, healthcare is funded through private and public insurance plans. Many Americans have to choose between employer-based or other private insurance plans that involve high out-of-pocket expenses through premiums, copayments, and deductibles [19]. Apart from the private insurance options available, the U.S. maintains special healthcare programs such as Medicare and Medicaid, supporting specific populations such as persons above 65, children under 21, and adults with dependent children, among others [19,20]. Most of the population in the U.S relies on private insurance plans, but the government ensures that at-risk groups can enroll in government-offered insurance plans such as Medicare and Medicaid [19,20]. Although there are multiple insurance options, approximately 27.5 million (8.5%) Americans were uninsured in 2018 [19]. This highlights critical gaps in coverage that create financial burdens for those unable to afford private insurance [19,21]. Private insurance continues to dominate the provision of insurance to approximately 67.3% of the insured population, with employer-based insurance covering approximately 55.1% [19]. On the other hand, Canada operates a single-payer, tax-funded model that provides universal healthcare to all (Medicare) that is managed at different levels, such as provincial and territorial administrative structures that facilitate the provision of services privately or publicly [20]. Canada’s Medicare provides universal health coverage without charge at the time of service, although this does not usually cover prescription drugs, dental health, or vision care [22].

There are some shared challenges between the U.S and Canada, although the two countries operate different healthcare systems. In both countries, there are ongoing challenges related to affordability, quality of service, and system accessibility [20]. A study on primary care crisis in Canada reported that one in six persons do not have personal physicians, and many struggle to find physicians when needed [23]. Additionally, results from a study by Kuile and others highlight that the Canadian healthcare system faces significant challenges in providing access to care for undocumented migrants and legal migrants who may struggle to fit within the existing provincial and federal health structures [24]. Unlike the focus on quicker access to specialists in the U.S., the Canadian system’s focus on universal access highlights a national emphasis on equitable coverage within a centralized cost model [20]. Both healthcare systems offer unique benefits to their clients, and both systems have unique and shared challenges and opportunities. This study observes that differences underscore distinct national priorities, stressing that the U.S. emphasizes choice and market competition while Canada prioritizes equitable access and cost control within a federally regulated framework.

## 3. Methods

### 3.1. Methodological Framework

This study adopted a theory-generating qualitative meta-synthesis approach. Originating from the insightful work of Finfgeld-Connett [17], this approach meticulously harnesses qualitative data from a spectrum of published research reports. By engaging in an inductive analytical process, the approach allows for extracting and integrating raw qualitative insights. Such a process is instrumental in constructing a context-rich body of knowledge, unveiling new perspectives crucial for informing decision-making processes and catalyzing meaningful action [25,26]. This methodology is pivotal in unearthing the experiences and viewpoints of IWDs as they navigate the complexities of healthcare systems. It also contributes significantly to understanding access barriers and facilitates the development of more inclusive and accessible healthcare services for one of the most understudied populations.

### 3.2. Database and Search Description

A literature search of peer-reviewed articles was conducted for articles published on immigrant and refugee access to healthcare in the United States and Canada. All articles considered included IWDs and were published in English. Most of the literature was original qualitative research, except for one literature review article that we included in the study because of its unique approach to autistic immigrant children with healthcare needs. Search words including barriers, access, immigrants, refugees, migrants, obstacles, challenges, medical care, disabilities, impairment, handicap, special needs, refugee health, immigrant families, and immigrant and refugee health were used to identify studies from PubMed, PsycINFO, Scopus, ProQuest, Google Scholar, Web of Science, and EBSCO.

### 3.3. Search Results, Review, and Selection Criteria

The search for studies was conducted in two stages. For the first-round search, PsycINFO, Scopus, PubMed, ProQuest, Google Scholar, Web of Science, and EBSCO databases were searched extensively. For the second round, a subject-based search was conducted to find articles from public health, sociological, and social work abstract databases. Zotero bibliographical software was used to collect, store, manage, remove duplicates, and facilitate the screening process. The two-stage search produced 751 studies, and one article was added from an additional source.

As demonstrated in Figure 1 below, a three-stage review was conducted to select the final sample for analysis. First, a title review of the 752 studies was conducted, and duplicates were removed. Then, 110 studies were selected for the second round of abstract review, and 24 articles were selected for the final full-text review. The full-text review produced 10 studies that met all the inclusion criteria (see Table 1) and were included for analysis.

The selection process of the meta-synthesis is reported under the Preferred Reporting Items for Systematic Reviews and Meta-Analyses (PRISMA) (Figure 1). 

### 3.4. Data Extraction, Analysis, and Quality

To ensure methodological rigor, we developed a study attributes table to highlight the methodological aspects that included the authors, study location, purpose, target population, sample size, and design method (see Table 2). The findings section of each selected study was copied and pasted into a separate document for analysis and review. Reflective notes and memos were systematically recorded during the initial review to discern patterns, differences, and relationships among the data [17]. Then, the findings from each study were uploaded into Nvivo14 software for coding and analysis using the grounded theory approach, which is in line with the theory-generating meta-synthesis [17]. After uploading the findings into NVivo 14 software, codes and themes were developed to guide coding and identify major themes and sub-themes across the selected articles. A codebook was created that highlighted the main categories and sub-categories together with appropriate definitions and examples. These reflective memos and categories were skillfully combined to generate meaning and establish relationships.

Besides the inclusion criteria set for the qualitative meta-synthesis (See Table 1), the studies were assessed using the Critical Appraisal Skills Programme UK (CASP) checklist (2018) to enhance quality and rigor (see Table 3). The selected studies met most of the CASP, including purpose, methodological appropriateness, recruitment, rigor, and clear presentation of the findings. Although researchers did not adequately address the relationship between researchers and participants, they reported employing linguistically and culturally competent interviewers and focus group discussion facilitators. No study was excluded based on the CASP assessment.

## 4. Results

### 4.1. Attributes of Sample Studies

The meta-synthesis analyzed ten qualitative studies that examined the challenges, facilitators, and other support mechanisms of accessing healthcare among IWDs in the United States and Canada. Five articles were from the U.S., and five were from Canada and were published between 2013 and 2024 with a combined total of 191 participants (n = 191). See Table 3 for detailed information on the attributes of the selected studies.

The analysis resulted in three major themes: (1) structural barriers, (2) cultural and personal barriers, and (3) facilitators of access to healthcare services. As presented below, several sub-themes were identified under each major theme across the ten studies.

### 4.2. Theme 1: Structural Barriers

#### 4.2.1. Bureaucracy and Complexity of the System

Immigrants with disabilities (IWDs) often face bureaucratic barriers in accessing healthcare and express frustration with the challenges faced in accessing healthcare [8,28,29,30]. In particular, those with an undocumented status could not qualify for insurance or Medicaid until they obtained legal status and thus were left without healthcare [33]. Immigrant caregivers of IWDs expressed frustration with the time it took to get support, irrespective of how long they had been in the U.S. [34]. Besides legal status, a lack of understanding of how the healthcare system works further constrained IWDs’ ability to access care [29]. The lack of understanding was particularly highlighted, whereby families were unsure where to go when they needed services. Even when IWDs found a way to get to healthcare providers, they were required to go through a long and arduous process of filling in the required documentation, making access to care extremely difficult [29].

#### 4.2.2. Healthcare Cost

Immigrant families, especially those with children with disabilities, struggle to meet the costs of healthcare in addition to meeting other competing needs [28,31,32]. The situation is complicated, especially when only one family member works and earns a small salary to support the entire family. Because of the financial and legal predicament, immigrant families having children with disabilities may not adequately support their needs and the needs of their dependents [31]. The need for support is more pronounced among persons with severe cognitive and functional disabilities [28,30,31]. The added economic strain is exacerbated by parents who take on a permanent caretaker role for persons with disabilities at the expense of their financial needs.

#### 4.2.3. Transportation Challenges

Immigrants with disabilities expressed facing challenges in accessing health facilities because of the problematic public transportation in many parts of the U.S. and Canada [28,35]. Studies highlight instances when immigrant families were located as far as 50 miles from healthcare facilities and missed medical appointments because of transportation challenges [28,35]. In other instances, there was a struggle to coordinate transportation and the availability of parents and caregivers to accompany their dependents to service providers [28,35]. Transportation issues were further exacerbated by the historic placement of immigrants in underserved communities with poor and aging transportation systems. Because of the challenges with transportation, many IWDs choose between going to work or spending the whole day commuting to health providers; usually, they choose to work [35].

#### 4.2.4. Information and Communication Gaps

Immigrants with disabilities experienced miscommunication with and among healthcare providers. They received inconsistent information from doctors, nurses, and other practitioners that hindered their healthcare decision-making [31]. Miscommunication between physicians or their offices and the pharmacies created challenges in receiving prescriptions and drugs correctly and on time [30,36]. Several instances of risking patients’ lives because of poor internal and external communication were cited in the studies. The danger of patients receiving multiple doses or missing out on medication was high amidst such confusion [29]. The findings showed deep systemic problems in healthcare institutions in keeping records updated and communicating clearly whenever a prescription was issued or canceled to avoid situations of having to take two or more prescriptions for the same treatment.

The communication challenges are also related to the knowledge gap on the providers’ side. This meta-synthesis highlights that some providers were oblivious to the available support opportunities for persons with disabilities, especially services for children with severe disabilities [29,36]. Service providers are not fully informed about the breadth of services and resources accessible to IWDs, including children. It was observed that agencies lack knowledge about the support options available for children with severe disabilities [29].

#### 4.2.5. Long Wait Times

Long waiting periods for receiving a diagnosis were identified as problematic for many IWDs, causing frustration in accessing healthcare. Where immigrant families managed to have insurance to access healthcare opportunities, it only lasted a maximum of eight months [30,31]. The issue of waiting periods extends to private diagnostic services, where some families turn due to the prolonged delays and bureaucracies within the public system. The time-consuming nature of accessing programs, characterized by extensive waiting lists, compounds the frustration. The lack of transparency and information about waitlist management further exacerbates these challenges [30,31]. This situation results in uncertainties about the timeline for accessing healthcare services. For instance, one case highlighted involves a mother who reported a four-year waiting period for services in north Toronto, contrasting with her own experience of over a year’s wait in her locality, illustrating inconsistency and lack of transparency [29]. Such prolonged waiting times can result in diminished efficacy of interventions like speech therapy, as they become less effective with the person’s advancing age.

#### 4.2.6. Lack of Integrated Services

Immigrants with disabilities struggle with having to deal with multiple referrals and a lack of an integrated healthcare system that would enable clients to receive multiple services under one roof [29,30,34]. Several first-to-contact providers recommended referrals, sometimes more than five times [34]. For example, a client had to be referred to at least 17 professionals in one year [35]. Families and caregivers reported frustrations due to many referrals that required changing from one service provider to another [30]. It was particularly challenging for IWD families to locate where services were offered, find transportation, and receive the needed services in a reasonable time. Referral challenges also arose because of the discordance between different health services, school systems, and the relevant state and federal departments [29,31].

### 4.3. Theme 2: Cultural and Personal Barriers

#### 4.3.1. Cultural Beliefs, Denial, and Trust

Some IWDs and their caregivers struggle to accept disability that hinders healthcare-seeking behavior [31,33,37]. In particular, immigrant parents initially refused to accept that their child had a disability until multiple medical examinations, diagnoses, and personnel changes confirmed a disability [31]. The delay in acceptance or outright denial of the diagnosis usually affects the evaluation process [34].

Understanding disabilities is linked to accepting results from diagnoses shaped by cultural leanings. Cultural perceptions of disabilities such as Autism Spectrum Disorder (ASD) are influenced by the prevailing norms of what is considered normal within that culture [33]. Diagnosis of disabilities among immigrants requires belief and understanding that the perceived disability is problematic. Findings from the synthesized studies affirm the connection between diagnosis and response influenced by one’s cultural perceptions [34,38]. Sritharan & Koola [33] observed that specific communities, including Latinos and Asians, believe that the family is responsible for the disabilities of their children. How immigrant caregivers understand disability in a culturally relevant context shapes their response to treatment. For example, Somali parents with autistic children expressed doubts about genetic explanations for autism [33]. This perspective is not unique to Somali parents but is shared among various groups who question genetic reasoning for autism and other disabilities [37]. This reflects a broader cultural challenge in the acceptance and understanding of genetic causes of disabilities that leads to a preference for non-conventional treatment methods, thus delaying response to care and treatment [35].

The denial is partly attributed to a lack of trust or confidence in service providers and the inability to reach a consensus between parents or caregivers to accept the diagnosis and treatment [31,34]. Some IWDs and their caregivers had concerns about providers’ inadequate awareness regarding IWDs. In most cases, several service providers dealing with immigrant health were not specialized in dealing with disabilities or they lacked basic disability training that would help to detect and support IWDs [8,12]. The problem is exacerbated by missing or inconsistent medical histories of IWDs before their resettlement in the U.S. or Canada. Consequently, this lack of insight often resulted in unsuitable recommendations for subsequent medical interventions, thereby hindering trust in healthcare and timely access to essential disability-related social services for immigrants [8,12].

#### 4.3.2. Stigma and Discrimination

Cultural understanding of disability influences IWDs’ perceptions and responses to the situation and treatment [31,33]. Some immigrant communities perceive disability as a punishment, curse, or the family’s fault. Some communities directly blame mothers for a disability in the family [30]. Because of cultural perceptions, blame, and ridicule, IWDs and their families may experience social isolation and exclusion [30]. Most IWDs and their caregivers delayed seeking health services due to concerns about stigmatization and public humiliation [32]. Stigma-induced fear was identified as a crucial factor for immigrant families of children with disabilities accessing healthcare [31]. In some cases, fear stems from a lack of information about disabilities and how to address their associated medical and social challenges [31].

#### 4.3.3. Awareness and Language Gaps

The cultural backgrounds and beliefs stemming from the home countries of IWDs can significantly impact their utilization of healthcare and educational services [29,31,33]. Some immigrants and their caregivers often do not proactively seek services, but rather express gratitude for whatever services are provided [29]. A prevalent lack of awareness about their fundamental rights to healthcare and education means IWDs and their caregivers are less likely to demand the necessary resources assertively [29,30]. A prevalent hesitation exists, with many considering access to services a privilege rather than an inherent right. This hesitancy frequently delays seeking assistance until situations reach a critical level [33].

Immigrants with disabilities and caregivers also expressed their frustration at the limited understanding of the various support options available to them [32]. There is a lack of awareness around the available options, such as reliable health facilities, information about health insurance, and disability-specific information [30,35]. Insufficient knowledge significantly hindered timely access to healthcare services and caused service disruptions extending beyond a year. For instance, locating services for children diagnosed with autism spectrum disorder (ASD) depended on the caregivers, requiring them to undertake extensive research on their own [35]. This challenge was compounded by the struggle to identify culturally competent healthcare providers who could respect cultural preferences and offer disability-specific, customized interventions [35].

Immigrants with disabilities also struggled to access healthcare because of their English language limitations and the unavailability of translation services [8,32]. Language is the main barrier between IWDs or their caregivers and health providers [37]. While some IWDs and caregivers understand English, many do not understand or comprehend the medical terminologies used [31]. Due to language and communication challenges, many IWDs give up on healthcare access [31].

#### 4.3.4. Lack of Social Support

Immigrants with disabilities struggle for adequate social support at various levels. The lack of social support is associated with the isolation of persons with disabilities and impacts their social well-being and overall performance in life [28,34]. As immigrants, the burden of having disabilities is amplified as they are socially disconnected from the community upon arrival in a new land [28]. The lack of community and social support predisposes IWDs, especially children, to challenges in dealing with and coping with anger issues [39]. Furthermore, several studies highlight caregivers feeling isolated because of their children’s or family member’s disability [28,30,31]. Some caregivers stated they could not find any support group for immigrant families with disabilities, although they have been waiting for as long as ten years [28,30,31]. Besides a lack of social support, IWDs are challenged by the conflicting roles of household members or caregivers that affect their access to healthcare. In most cases, despite mothers playing a 24/7 caregiving role, fathers make healthcare decisions. This disparity in roles is problematic because fathers may not be realistic in their expectations for their children’s behavior, and thus challenges for the children experiencing disability remain unaddressed [29,30].

### 4.4. Theme 3: Facilitators of Access to Healthcare

#### 4.4.1. Immediate Family and Relatives

Immediate family members and relatives play a central role in facilitating IWDs’ access to healthcare. They take on responsibilities as caregivers and act as protectors from the abuse and exploitation of those with disabilities [35]. They serve as translators, provide transportation, represent and advocate for IWDs in healthcare settings, and process documentation [31]. Thus, having supportive parents and relatives is crucial for IWDs to access the healthcare support they need. This was strengthened through networking with other families of IWDs and sharing their experiences and how they successfully advocated for their people’s needs [29].

#### 4.4.2. Community Health Centers

When immigrants do not have health insurance or access to advanced public healthcare or private healthcare providers, they seek help from community health centers and rehabilitation centers. These institutions, although not widely available, facilitated affordable care and IWDs’ entry into the healthcare system [34]. Early engagement with community health centers was crucial as the waiting times could be longer and resources are limited [32].

#### 4.4.3. Social Workers’ Support

Social workers play a critical role in providing support for IWDs in accessing healthcare services. Through specialized or intensive case management and home visits, which include tailored social and medical support for specific mental or physical health challenges, social workers facilitate care for IWDs [28,35]. For example, social workers helped children struggling to read three-letter words, leading to a recommendation for school assessment, illustrating the customization and ongoing support [31,32]. Caregivers of IWDs reported that they find it easier to work with social workers in facilitating care because of their professional and human qualities, such as competence, determination, and kindness [31,32]. This support extended from tangible interventions like securing assessments to emotional and informational support facilitated through regular visits and ensuring alignment with family goals.

## 5. Discussion

This qualitative meta-synthesis highlighted the barriers and facilitators of healthcare access among IWDs in the United States and Canada. Based on the synthesized data, we argue that these barriers and facilitators impact access to healthcare within both the United States and Canadian contexts, irrespective of the differences in the respective healthcare systems. A diagram that shows the various barriers at different levels and facilitators of access to healthcare is presented in Figure 2. As demonstrated in the model, IWDs experience challenges in accessing healthcare mainly on two levels: (a) structural and (b) cultural and personal barriers. The meta-synthesis revealed that although the identified barriers seem to be shared among the general immigrant population, each barrier has unique aspects that apply to IWDs. The impacts of each of the barriers will be heightened when experienced by IWDs because of their disadvantaged position among the general immigrant population. Hence, this study provides insights into the unique needs of IWDs in accessing healthcare and developing tailored interventions to improve access for this population.

Structural or system-level barriers were the most reported or dominant hindrances to healthcare access for IWDs in the United States and Canada. The structural barriers were further categorized into six different types, as depicted in Figure 2. The first five barriers were repeatedly reported as barriers to access to healthcare in the general immigrant population, while the last barrier is unique to IWDs. The findings align with previous studies that reported the prevalence of lengthy bureaucracy and complexity in the healthcare system as one of the main hindrances to healthcare access [32,40]. A previous study on the experiences of immigrant parents of children with disabilities navigating health and rehabilitation services in Norway underscores that immigrants are overburdened by the paperwork and legal documents required by healthcare providers to access healthcare [41]. Attributable to factors such as immigrant status, discriminatory policies, or biases, IWDs are often required to complete extensive paperwork or disclose sensitive personal information [41]. A previous review of the literature on the barriers faced by immigrant families of children with intellectual and developmental disabilities reported that such complexities and procedures disproportionately affect those who are undocumented and with disabilities [42]. As a result, IWDs and their caregivers may experience frustration and perceive risks in seeking healthcare, often delaying access to services until their health conditions become severe [43]. The insurance system with varied requirements and specifications about inside and outside network providers, rules around covered and excluded types of care, and fragmented specialized care is overwhelming for immigrants [43,44]. The lack of integrated services that encompass various specialties and disability services worsened the challenges experienced by IWDs. The unavailability of disability-oriented and specialized services in mainstream healthcare results in long wait times and in the ability to get appointments and timely services for IWDs.

The impacts of bureaucracy and complexity are exacerbated by the lack of adequate and clear information, communication gaps in the healthcare system, and knowledge and linguistic challenges on the IWDs’ side. Communication barriers are often heightened by language differences and a lack of culturally competent care, further alienating IWDs and their families from the healthcare system. A lack of translation services, especially those that accommodate the needs of IWDs and that involve in-person non-verbal communication, is one of the main obstacles in care provision for IWDs. A lack of customized information and IWDs’ inability to understand medical language are also impeding factors [45]. The findings emphasize the critical need for language-appropriate services and cultural sensitivity training for healthcare providers to improve healthcare access and outcomes for immigrant populations [46].

The high cost of healthcare has generally been documented as the primary barrier for immigrants in the synthesized studies [30,31]. However, the problem disproportionately affects IWDs, as they are further discriminated against in the workforce environment and have limited employment opportunities that allow them to obtain employment-based insurance and be financially fit to afford healthcare [29]. Like the cost of care, transportation challenges uniquely affect IWDs because of accessibility challenges and various restrictions in obtaining driver’s licenses among IWDs. Especially in the U.S., where public transportation is limited, IWDs can experience significant challenges in getting to and from healthcare services, resulting in delays and cancelation of services [30,35]. The structural and systemic barriers are compounded by cultural and personal factors such as stigma and discrimination, beliefs about disability, and denial of disability. The discrimination IWDs experience is two-sided. First, they face discrimination for being immigrants, usually perpetrated by non-immigrants or systems in the host country. Second, they get discriminated against for their disabilities within their own immigrant community and the larger population [31]. As presented in the findings and the model (Figure 2), the stigma and discrimination against disability is deep-rooted in most immigrant cultures, creating social isolation and deteriorating social support for IWDs and their families [31]. The findings highlight a clash between traditional beliefs held by immigrant families and the biomedical model predominantly practiced in the United States and Canadian healthcare settings [47]. This cultural discordance can delay the seeking of care and acceptance of diagnoses, supporting the conclusions drawn by Cabieses and others [48] regarding the impact of cultural stigma on healthcare access among immigrant populations.

Although outweighed by the barriers, the findings revealed three facilitators of access to healthcare among IWDs. As support from the healthcare system, as well as the larger community, is limited, IWDs mainly rely on their immediate family members and close relatives to access and utilize healthcare. Due to the stigma toward disability, the nature of social support received by IWDs is very limited and tied to immediate family members compared to the general immigrant population, who enjoy support from the larger immigrant community in accessing healthcare [31,40].

Despite their lack of adequate resources and specialized care, community health centers remain the main source of medical support for IWDs. Previous research carried out among Latino immigrants in the U.S. highlighted that community health centers are known to provide affordable care for immigrant communities through income-based sliding-scale payment systems and sometimes free basic health services [49]. Support from social workers was the last, but a unique facilitator emerged in this meta-synthesis. Various systematic reviews on facilitators of access to healthcare among the general immigrant population did not report social workers’ facilitative role in promoting access [40]. The findings demonstrate the unique position social workers hold in understanding the medical, social, cultural, and economic facets of disability and their ability to link IWDs to various services using case management processes.

### 5.1. Policy and Practice Implications

The results of this meta-synthesis reveal significant insights into the barriers and facilitators to healthcare access for IWDs in the United States and Canada. These findings have several implications for policy and practice, which are crucial for improving healthcare access among immigrants, especially those with disabilities.

To address the lengthy bureaucratic hurdles and ease the complexity of healthcare, it is important to initiate policy reform that reduces the amount of paperwork and documentation requirements for IWDs, especially those asking for sensitive immigration-related information. Providing customized health information that targets IWDs and multilingual services and improving internal communications to provide consistent information for service users is critical. To ease the burden of navigating fragmented healthcare services, creating a one-stop integrated healthcare service for IWDs where their needs are addressed is important. Policies should aim to streamline referral processes, promote effective inter-agency collaboration, and develop integrated service models to reduce the challenges of accessing healthcare from multiple sites [50]. Similarly, streamlined, integrated technology, such as one-stop and multilingual mobile apps, could enhance healthcare access for IWDs.

Regarding the identified socioeconomic barriers, addressing these barriers should involve providing support for families of persons with disabilities, especially where there is a single or no breadwinner [30,32]. To address financial challenges and the cost of care, it is critical to strengthen and expand community healthcare centers that provide income-based affordable care for IWDs and their families. Revisiting restrictive policies, such as the Public Charge rule and various state-based regulations, and providing resources such as subsidies for healthcare costs, increased economic assistance, and supportive services for IWDs, is essential [51]. Furthermore, addressing transportation challenges through better coordination of services and providing subsidies or dedicated transport services for healthcare appointments can significantly improve access.

Multiple referrals and a lack of integrated services were considered prohibitive to healthcare access [50]. These underscore the need for a more accessible and coordinated healthcare system. Policies should aim to streamline referral processes, promote effective inter-agency collaboration, and develop integrated service models to reduce the challenges of accessing healthcare from multiple sites. This will ensure a better way for immigrants to navigate the healthcare system and receive healthcare support.

Given the significant role of cultural beliefs in healthcare access, it is imperative to establish a culturally competent healthcare system. This requires training healthcare professionals on immigrant cultural nuances, offering multilingual services, and embedding cultural sensitivity into healthcare practices to enhance communication, trust, and adherence to medical advice [52,53]. Furthermore, addressing ignorance and denial around disabilities through educational programs for both immigrants and healthcare providers is crucial, focusing on raising awareness about disabilities, support options, and healthcare rights [54]. This should also involve educating providers on the diverse needs of IWDs and the significance of culturally sensitive care, underscoring a holistic approach to improving healthcare access and outcomes for immigrant populations [54]. The findings from this meta-synthesis call for comprehensive policy changes and targeted interventions to address the multifaceted barriers to healthcare access for IWDs in the United States and Canada. Addressing these barriers will enhance the possibilities of moving towards a more equitable and accessible healthcare system for all individuals, regardless of their status.

### 5.2. Limitations

#### 5.2.1. Limited Literature to Synthesize

Qualitative research focusing on IWDs and healthcare access in the U.S. and Canada is limited, affecting our ability to identify and synthesize data at a larger level. Hence, only ten studies were included in the synthesis; two are from one larger study [36,37]. To address this, a detailed and thorough analysis led to the development of a comprehensive model of barriers and facilitators of access. This model could be enriched and modified with further research.

#### 5.2.2. Contextual Limitations

This study provides comprehensive insights into the unique healthcare needs of IWDs in the United States and Canada. The coverage of two countries with different healthcare systems was occasioned by a lack of an adequate number of publications that met the inclusion criteria in the individual countries as each has distinct immigration, healthcare, and disability support policies, albeit with some similarities; we acknowledge that aggregating the findings from both countries may obscure crucial contextual differences. In applying the thematic conclusions, it is imperative to bear in mind the contextual differences between the two healthcare systems. We recommend that future meta-syntheses (with more research and data availability) consider focusing on one country at a time to enhance the contextual specificity of findings, thereby facilitating the development of more targeted, country-specific recommendations.

#### 5.2.3. Demographic Limitations

The sample studies cover a wide demographic range, encompassing individuals of varying ages, as well as both physical and mental disabilities, without sufficient stratification by these subcategories. Further research is needed in the two countries to expand knowledge based on the experience of IWDs in their search for dignified healthcare access. Future research should explore how healthcare access barriers and facilitators vary by demographic and disability type, enhancing the precision of tailored intervention strategies.

#### 5.2.4. The Interruption of COVID-19

This study covered ten years without separately addressing the COVID-19 period. We postulate that because of the extensive impact of COVID-19, future studies should attempt to address specifically how COVID-19 affected healthcare access for vulnerable populations such as IWDs because the pandemic might have intensified the challenges faced by IWDs in accessing healthcare.

## 6. Conclusions

This meta-synthesis reveals a complex intersection of structural, socioeconomic, cultural, and communicative barriers that significantly impact healthcare access for IWDs in the U.S. and Canada. Although the findings relate to two different welfare systems, there are some shared and unique challenges that impact healthcare access for IWDs. Structural issues—such as the complexity of healthcare systems, high costs, and transportation difficulties—create formidable barriers that are exacerbated by socio-economic constraints and systemic limitations, especially for vulnerable groups like IWDs. The cultural and personal barriers identified, including stigma, denial, and a lack of trust, further complicate healthcare access for IWDs. These complex and intersecting barriers present a challenge to the highly fragmented, complex, inaccessible, and poorly coordinated healthcare systems. In addition, the lack of a culturally competent and diverse healthcare workforce will make handling the identified challenges difficult. However, we also acknowledge that these challenges create valuable opportunities to explore innovative, integrated service models that can streamline access and minimize the bureaucratic and economic burdens on IWDs.

We strongly recommend a policy and practice reform that reduces the amount of paperwork and documentation requirements and provides multilingual services to IWDs. Reform efforts should include better-aligned provider training on cultural competence, improved inter-agency collaboration, and more robust social support networks within immigrant communities and the healthcare system. The development and utilization of technological products such as a one-stop multi-lingual mobile app is highly recommended to streamline information and improve healthcare utilization among IWDs. It is also crucial to provide economic support to IWDs through employment opportunities to boost their financial capacity to afford healthcare and transportation costs. Expanding affordable community-based health clinics can significantly improve access among IWDs. This will also foster more inclusive, accessible, and responsive healthcare environments. Expanding disability awareness education among immigrant communities is critical for tackling stigma and discrimination and promote inclusive living and access to healthcare for IWDs. To conclude, improving access to healthcare for IWDs requires a multi-sectoral and integrated approach that involves policy and practice reform in the healthcare system, as well as the active participation of the immigrant community through education and community-based health centers.

## Figures and Tables

**Figure 1 healthcare-13-00313-f001:**
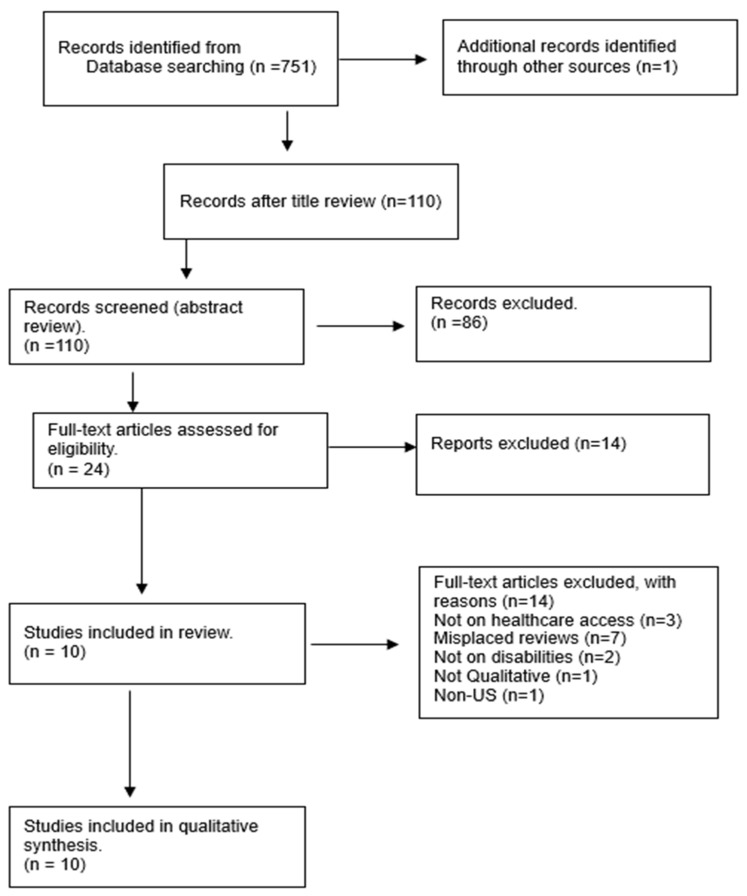
Review and selection process of studies. Note. Adapted from “PRISMA Flow Diagram” by [27].

**Figure 2 healthcare-13-00313-f002:**
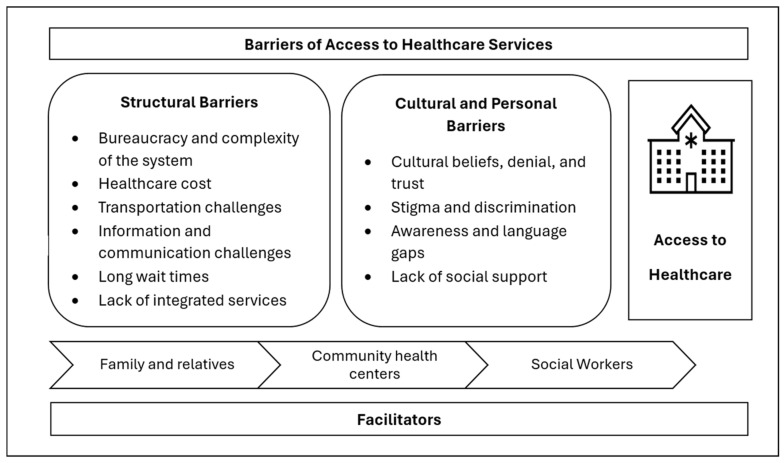
Model of barriers and facilitators to healthcare access among IWDs.

**Table 1 healthcare-13-00313-t001:** Inclusion and exclusion criteria.

Inclusion Criteria	Exclusion Criteria	Explanation/Rationale
Target population and focus: IWDs	Non-immigrant population Not related to disability and access to healthcare	The focus of this qualitative meta-synthesis was to explore the experiences and perspectives of IWDs
Studies performed in the U.S. and Canada		To maintain contextual relevance
Qualitative approach	Quantitative or mixed study	The project is a qualitative meta-synthesis
Articles written in English		The research team spoke English
Published between 2013 and 2024		To focus on findings within the last ten years
Research reports of original research	Literature, systematic reviews, and commentaries	To understand and synthesize actual real-life experiences and perspectives of IWDs

**Table 2 healthcare-13-00313-t002:** Attributes of Sample Studies.

Author	Location	Purpose	Target Population	Sample	Design/Method
[28]	Baltimore	“The purpose of the scoping study was to understand the experiences of refugees with disabilities” (p. 189).	Refugees and migrants and their families	6	Snowball sampling, interviews, and semi-structured interviews.
[29]	Toronto	“The objectives of the Mothers Project were to understand the social support experiences and service needs of immigrant mothers of children with disabilities, and to investigate service providers’ perspectives on the challenges faced by immigrant mothers in accessing social support and services” (p. 1841).	Health service providers	27	Single-stage purposive sampling strategy and in-depth interviews.
[30]	Canada	“This paper presents findings from a larger study, Mothers Project, which explored the perspectives of mothers and service providers regarding social support needs, challenges and experiences of immigrant mothers of children with disabilities in Toronto, Canada” (p. 242).	Immigrant mothers of children with autism	21	Interviews.
[31]	Quebec	“The objectives of the present study were as follows: to (1) document the obstacles experienced by immigrant families in obtaining an ASD diagnosis for their child, (2) document the factors that facilitated their access to a diagnosis, (3) identify prevailing attitudes toward ASD in participants’ culture of origin, and (4) record the advice that participants would give to other immigrant families in their own trajectory to obtain a diagnosis” (p. 521).	Immigrant families	24	Semi-structured interviews using a sociodemographic questionnaire.
[32]	Ontario	“To examine the barriers and facilitators to health and social service access and utilization for immigrant parents raising a child with a physical disability, in order to understand their specific needs and experiences of care” (p. 135).	First-generation immigrants from Asia, Africa, and the Caribbean	5	Semi-structured interviews were analyzed using grounded theory.
[8]	Chicago	“There is a need to explore this population’s access to appropriate healthcare services in order to identify service disparities and improve interventions” (p. 733).	Disabled and chronically ill refugees	18	Community-based participatory research, semi-structured key informant interviews, and community meetings.
[33]	Toronto	“The objectives of this article are (1) to understand the barriers that immigrant families of children with ASD face and (2) to describe a culturally sensitive program model to address the barriers and provide targeted and accessible resources to these immigrant families” (p. 53).	Immigrants with ASD	21	Literature review.
[34]	U.S.	“To understand the ASD diagnosis and treatment pathways for U.S. families” (p. 1017).	Mexican-heritage mothers of children with ASD	38	Multiple-case design.
[35]	U.S.	“This qualitative study seeks to discover the particular challenges that IWDs face when accessing health care and the facilitating factors that assist them in this process” (p. S64).	Immigration from three communities	9	Purposive sampling, Multicase study design, interviews, and participant observation.
[36]	U.S.	“This study aimed to explore how Asian immigrant parents of CSHCNs view their child’s healthcare access, quality, and utilization” (p. 251).	Children with special healthcare needs	22	Semi-structured, standardized interview guided by grounded theory analysis.

**Table 3 healthcare-13-00313-t003:** CASP quality assessment for selected studies.

Checklist	Study									
	[28]	[29]	[30]	[31]	[32]	[12]	[33]	[34]	[35]	[36]
Was there a clear statement of the aims of the research?	Yes	Yes	Yes	Yes	Yes	Yes	Yes	Yes	Yes	Yes
Is a qualitative methodology appropriate?	Yes	Yes	Yes	Yes	Yes	Yes	Yes	Yes	Yes	Yes
Was the research design appropriate to address the aims of the research?	Yes	Yes	Yes	Yes	Yes	Yes	Not applicable	Yes	Yes	Yes
Was the recruitment strategy appropriate to the aims of the research?	Yes	Yes	Yes	Yes	Yes	Yes	Yes	Yes	Yes	Yes
Was the data collected in a way that addressed the research issue?	Yes	Yes	Yes	Yes	Yes	Yes	Yes	Yes	Yes	Yes
Has the relationship between the researcher and participants been adequately considered?	Yes	Yes	Yes	Yes	Yes	Yes	Yes	Yes	Yes	Yes
Have ethical issues been taken into consideration?	Yes	Yes	Yes	Yes	Yes	Yes	Yes	Yes	Yes	Yes
Was the data analysis sufficiently rigorous?	Yes	Yes	Yes	Yes	Yes	Yes	Yes	Yes	Yes	Yes
Is there a clear statement of findings?	Yes	Yes	Yes	Yes	Yes	Yes	Yes	Yes	Yes	Yes
How valuable is the research?	Very valuable	Very valuable	Very valuable	Very valuable	Very valuable	Very valuable	Very valuable	Very valuable	Very valuable	Very valuable

## Data Availability

Not applicable to qualitative meta-synthesis.

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
