# Peer review of "Barriers and Facilitators of Access to Healthcare Among Immigrants with Disabilities: A Qualitative Meta-Synthesis"

_healthcare, 2025, doi:10.3390/healthcare13030313_

Round 1
Reviewer 1 Report
Comments and Suggestions for Authors
This article includes a thorough articulation of the barriers facing Immigrants with Disabilities (IWDs) attempting to access healthcare in the US and Canada through a comprehensive review of the literature. It employs a novel and comprehensive theory-generating meta-synthesis strategy articulated by Finfgeld-Connett (2018). The authors identified 752 studies that met their criteria, then explained with detailed models the systematic process why which they narrowed down their analysis to 110 studies selected for abstract review, 24 studies for full text review, and a final 10 studies that met all their inclusion criteria that formed the basis of the inductive analysis reported in the article. Tables and diagrams are offered to visually represent this process.
The authors produced a model which represent their findings related to “Barriers of Access to Healthcare Services.” The model is supported with a comprehensive narrative report of the varied yet related structural, cultural, and personal barriers that immigrants with disabilities face when trying to access healthcare in the US and Canada. The authors also include a discussion of positive mediating factors that facilitate access to healthcare (family/friends, health centers, and social workers). The authors draw on the model to make suggestions on how to improve access to healthcare at structural and policy levels. They conclude by noting that future research should focus on one context (i.e. US or Canada), but that more research is needed first.
I found this article to be an important contribution to the literature on healthcare and immigration, while serving as an excellent model for instructional use completing a meta-analysis / lit review. It is well written, cited, supported empirically, and includes an acknowledgement of the limitation of combining two distinct healthcare contexts (US and Canada).
More could be done to articulate the differences between the structural and cultural contexts of the US vs Canada healthcare systems to avoid conflating them. What does each system appear to do well? What can they learn from each other? The meta-analysis should be able to identify these differences.
Author Response
|
Reviewer comment |
Response |
|
This article includes a thorough articulation of the barriers facing Immigrants with Disabilities (IWDs) attempting to access healthcare in the US and Canada through a comprehensive review of the literature. It employs a novel and comprehensive theory-generating meta-synthesis strategy articulated by Finfgeld-Connett (2018). The authors identified 752 studies that met their criteria, then explained with detailed models the systematic process why which they narrowed down their analysis to 110 studies selected for abstract review, 24 studies for full text review, and a final 10 studies that met all their inclusion criteria that formed the basis of the inductive analysis reported in the article. Tables and diagrams are offered to visually represent this process.
|
We appreciate your thoughtful and encouraging reviews |
|
The authors produced a model which represent their findings related to “Barriers of Access to Healthcare Services.” The model is supported with a comprehensive narrative report of the varied yet related structural, cultural, and personal barriers that immigrants with disabilities face when trying to access healthcare in the US and Canada. The authors also include a discussion of positive mediating factors that facilitate access to healthcare (family/friends, health centers, and social workers). The authors draw on the model to make suggestions on how to improve access to healthcare at structural and policy levels. They conclude by noting that future research should focus on one context (i.e. US or Canada), but that more research is needed first.
|
Thank you for these constructive reviews |
|
I found this article to be an important contribution to the literature on healthcare and immigration, while serving as an excellent model for instructional use completing a meta-analysis / lit review. It is well written, cited, supported empirically, and includes an acknowledgement of the limitation of combining two distinct healthcare contexts (US and Canada).
|
We appreciate your feedback. |
|
More could be done to articulate the differences between the structural and cultural contexts of the US vs. Canadian healthcare systems to avoid conflating them. What does each system appear to do well? What can they learn from each other? The meta-analysis should be able to identify these differences
|
To ensure that we address this. We have added a short section specifically highlighting the context and the differences and stating that although the two countries operate different healthcare systems, some challenges are faced in both countries. |
Reviewer 2 Report
Comments and Suggestions for Authors
The use of standardized literature review processes, such as those mentioned, is positively evaluated. Likewise, the reviewer fully agrees with the statement: “Addressing these barriers will enhance the possibilities of moving towards a more equitable and accessible healthcare system for all individuals, regardless of their status.”
There are claims that require citation, as they are not necessarily accurate, such as: “The term immigrant represents people who move from one country to another with the intention of residing there permanently.”
When analyzing studies published between 2013 and 2023, it would have been necessary to contextualize them in terms of whether they precede, coincide with, or immediately follow the COVID-19 pandemic, a determining factor that is not even mentioned in the manuscript. It is highly likely that there are greater barriers or deficiencies in the system during and after this period.
Methodologically, it is not appropriate to combine the U.S. and Canada in the analysis, considering that they have different immigration, healthcare, and disability support systems. In fact, the 10 studies reviewed report data exclusively from one of the two countries. The authors are aware of this limitation, but it is not adequately addressed: “to at least focus on contextually closer geographies. When more qualitative research is available, we suggest that future meta-synthesis studies focus on a single country or immigration and healthcare context to develop more practical and contextually sound interventions and recommendations.”
This limitation significantly affects the results, as the thematic analysis in this section does not distinguish between what is applicable to the U.S. and Canada. In fact, claims are made that apply to both countries based on other studies, when in reality these studies are based solely on one of the two countries: “Immigrant caregivers of IWDs expressed frustration with the time it took to get support, irrespective of how long they had been in the U.S. or Canada (Cohen et al., 2023)” and “Immigrants with disabilities expressed facing challenges in accessing health facilities because of the problematic public transportation in many parts of the U.S. and Canada (Bogenschutz, 2014; Hamidi & Karachiwalla, 2022).”
Moreover, the healthcare system specifically designed for people with disabilities and for migrant reception in each country is not synthesized in the introduction or in any other section. In the humble opinion of the reviewer, these limitations are significant enough to reconsider the manuscript’s suitability for a journal of Healthcare’s impact.
In some instances, "U.S." is written, while in others it is "US."
A parenthetical citation should be reviewed: “Because of the financial and legal predicament, immigrant families having children with disabilities may not adequately support their needs and the needs of their dependents [33].”
It is true that a proper review of the state of the art, applied to the reality of each country and not jointly, can lead to policy recommendations. However, the claim that “this study provides insights into the unique needs of IWDs in accessing healthcare and developing tailored interventions to improve access for this population” seems too ambitious, as the interventions cannot be very tailored if they are directed equally at children and the elderly, nor is there a distinction between various physical or mental disabilities.
In section 4.1, proposals for improvement are made, but since they are based on studies from two different countries, three of which are 10 years old, these aspects should at least be considered as limitations when assessing the appropriateness of these proposals: how can the authors ensure the current relevance of these recommendations?
Author Response
|
Reviewer comments |
Responses |
|
The use of standardized literature review processes, such as those mentioned, is positively evaluated. Likewise, the reviewer fully agrees with the statement: “Addressing these barriers will enhance the possibilities of moving towards a more equitable and accessible healthcare system for all individuals, regardless of their status.”
|
Thank you so much for your feedback. We appreciate your critical and supportive review. |
|
There are claims that require citation, as they are not necessarily accurate, such as: “The term immigrant represents people who move from one country to another with the intention of residing there permanently.”
|
This has been addressed and a less restrictive definition adopted. |
|
When analyzing studies published between 2013 and 2023, it would have been necessary to contextualize them in terms of whether they precede, coincide with, or immediately follow the COVID-19 pandemic, a determining factor that is not even mentioned in the manuscript. It is highly likely that there are greater barriers or deficiencies in the system during and after this period.
|
This is noted and has been addressed as a limitation |
|
Methodologically, it is not appropriate to combine the U.S. and Canada in the analysis, considering that they have different immigration, healthcare, and disability support systems. In fact, the 10 studies reviewed report data exclusively from one of the two countries. The authors are aware of this limitation, but it is not adequately addressed: “to at least focus on contextually closer geographies. When more qualitative research is available, we suggest that future meta-synthesis studies focus on a single country or immigration and healthcare context to develop more practical and contextually sound interventions and recommendations.”
|
It is a limitation. We have, to the extent possible, given examples from the two countries used. We acknowledge that the two systems are different, while at the same time, we appreciate that some challenges and problems are experienced in both countries. We have highlighted this in our added paragraph highlighting the two healthcare systems nature and functioning. |
|
This limitation significantly affects the results, as the thematic analysis in this section does not distinguish between what is applicable to the U.S. and Canada. In fact, claims are made that apply to both countries based on other studies, when in reality these studies are based solely on one of the two countries: “Immigrant caregivers of IWDs expressed frustration with the time it took to get support, irrespective of how long they had been in the U.S. or Canada (Cohen et al., 2023)” and “Immigrants with disabilities expressed facing challenges in accessing health facilities because of the problematic public transportation in many parts of the U.S. and Canada (Bogenschutz, 2014; Hamidi & Karachiwalla, 2022).”
|
We have also clarified to situate the claim against the particular study. Those areas in which similar challenges are mentioned in both countries have been maintained e.g. transportation |
|
Moreover, the healthcare system specifically designed for people with disabilities and for migrant reception in each country is not synthesized in the introduction or in any other section. In the humble opinion of the reviewer, these limitations are significant enough to reconsider the manuscript’s suitability for a journal of Healthcare’s impact.
|
We have added a paragraph with a full section on the U.S and Canadian healthcare systems |
|
In some instances, "U.S." is written, while in others it is "US."
|
This has been addressed by consistently using U.S. |
|
A parenthetical citation should be reviewed: “Because of the financial and legal predicament, immigrant families having children with disabilities may not adequately support their needs and the needs of their dependents [33].”
|
This has been replaced with the author (Rivard et al., 2019) |
|
It is true that a proper review of the state of the art, applied to the reality of each country and not jointly, can lead to policy recommendations. However, the claim that “this study provides insights into the unique needs of IWDs in accessing healthcare and developing tailored interventions to improve access for this population” seems too ambitious, as the interventions cannot be very tailored if they are directed equally at children and the elderly, nor is there a distinction between various physical or mental disabilities.
|
We have adjusted this with a revised conclusion. |
|
In section 4.1, proposals for improvement are made, but since they are based on studies from two different countries, three of which are 10 years old, these aspects should at least be considered as limitations when assessing the appropriateness of these proposals: how can the authors ensure the current relevance of these recommendations?
|
This has been noted as a limitation due to limited studies in this area. |
Reviewer 3 Report
Comments and Suggestions for Authors
Line NO.
7-Include (IWD) here so readers are familiar later with the acronym later.
26, 36 distinguish between immigrants and refugees if there is a difference, or are they the same?
Specify whether the immigrants are legal or illegal migrants.
60 - The phrase due to refers to money and dates, best to use because of instead.
Tables in pages 1 and 2 would help.
98 - delete the word lived - redundant with experiences.
124 - Why not write a review of literature conducted ... instead of one member?
142 to 148 - Briefly introduce the tables, then after the table, explain the contents of the table. Do the same for any figures presented as well.
157 - Introduce table and explain contents after table.
174 - Last paragraph in 2.4. Eliminate the word Furthermore... Begin with Besides ... or In addition to, ...
192 -summarized Table 3 contents.
257 single digits should be written, 8 = (eight).
276 - Use because of or the resolt of instead of due to which refers to dates and money due.
422 - Should this be Table 4?
484 - Write Cabieses and others in the text.
489 - write, "Because of" ... instead of Due to...
References: APA formatting, In several references, the article titles from journals should be in lower case only, except first word, and the journal title should be in upper case and italics, e.g., 604.
Author Response
|
Reviewer Comments |
Responses |
|
7-Include (IWD) here so readers are familiar later with the acronym later.
|
IDWs added on line 7 |
|
26, 36 distinguish between immigrants and refugees if there is a difference, or are they the same? Specify whether the immigrants are legal or illegal migrants.
|
This has been rectified |
|
60 - The phrase due to refers to money and dates, best to use because of instead. Tables in pages 1 and 2 would help |
Replaced with because of |
|
98 - delete the word lived - redundant with experiences.
|
Word deleted |
|
124 - Why not write a review of literature conducted ... instead of one member?
|
Recommendation addressed: thank you |
|
142 to 148 - Briefly introduce the tables, then after the table, explain the contents of the table. Do the same for any figures presented as well.
|
These are introduced in previous lines eg line 146 for table 162, table 3, two is addressed in line 176 |
|
.157 - Introduce table and explain contents after table.
|
Addressed in line 172 through 180 |
|
174 - Last paragraph in 2.4. Eliminate the word Furthermore... Begin with Besides ... or In addition to, ...
|
We have addressed and improved the last paragraph to address this concern as well. |
|
192 -summarized Table 3 contents.
|
These have already been summarized and references to them have been made in the write-up |
|
257 single digits should be written, 8 = (eight).
|
Written as eight |
|
276 - Use because of or the result of instead of due to which refers to dates and money due.
|
Replaced with “because of” |
|
422 - Should this be Table 4?
|
Table 4 header added…. |
|
484 - Write Cabieses and others in the text.
|
“and others” has been added to the text |
|
489 - write, "Because of" ... instead of Due to...
|
addressed |
|
References: APA formatting, In several references, the article titles from journals should be in lower case only, except first word, and the journal title should be in upper case and italics, e.g., 604.
|
Addressed |
Reviewer 4 Report
Comments and Suggestions for Authors
The proposed study is an intriguing and well-structured endeavour. The text is easy to understand and follows a logical sequence. The methodology employed is suitable for this type of research. The results are presented in a clear and concise manner. The discussion is well-founded and provides a comprehensive overview of the contrasting information. The only shortcoming is that the conclusion is too brief. A work of this depth should include a greater number of conclusions. Indeed, further elaboration on the aforementioned intersections is required. What challenges do they pose for future research or for the management of the problem at hand? Conversely, the barriers provide an opportunity for health system transformation, but no conclusions have been drawn. I am confident that this is not the case.
Author Response
|
Reviewer comments |
Responses |
|
The proposed study is an intriguing and well-structured endeavor. The text is easy to understand and follows a logical sequence. |
Thank you so much for your thoughtful comments. We greatly appreciate your time and encouragement. |
|
The methodology employed is suitable for this type of research. The results are presented in a clear and concise manner. |
|
|
The discussion is well-founded and provides a comprehensive overview of the contrasting information. |
|
|
The only shortcoming is that the conclusion is too brief. A work of this depth should include a greater number of conclusions. Indeed, further elaboration on the aforementioned intersections is required. What challenges do they pose for future research or for the management of the problem at hand? Conversely, the barriers provide an opportunity for health system transformation, but no conclusions have been drawn. I am confident that this is not the case.
|
We have addressed the paragraph with an expanded explanation. |
